# Empowering Patients to Self-Manage Common Infections: Qualitative Study Informing the Development of an Evidence-Based Patient Information Leaflet

**DOI:** 10.3390/antibiotics10091113

**Published:** 2021-09-15

**Authors:** Catherine V. Hayes, Bláthnaid Mahon, Eirwen Sides, Rosie Allison, Donna M. Lecky, Cliodna A. M. McNulty

**Affiliations:** Primary Care and Interventions Unit, Public Health England, Gloucester GL1 1DQ, UK; Blathnaid.mahon@phe.gov.uk (B.M.); Eirwen.sides@phe.gov.uk (E.S.); Rosie.allison@phe.gov.uk (R.A.); Donna.lecky@phe.gov.uk (D.M.L.); Cliodna.mcnulty@phe.gov.uk (C.A.M.M.)

**Keywords:** primary healthcare, general practice, community pharmacy, antibiotics, qualitative study, patient attitudes, self-care, questionnaire, behavioural science

## Abstract

Common self-limiting infections can be self-managed by patients, potentially reducing consultations and unnecessary antibiotic use. This qualitative study informed by the Theoretical Domains Framework (TDF) aimed to explore healthcare professionals’ (HCPs) and patients’ needs on provision of self-care and safety-netting advice for common infections. Twenty-seven patients and seven HCPs participated in semi-structured focus groups (FGs) and interviews. An information leaflet was iteratively developed and reviewed by participants in interviews and FGs, and an additional 5 HCPs, and 25 patients (identifying from minority ethnic groups) via online questionnaires. Qualitative data were analysed thematically, double-coded, and mapped to the TDF. Participants required information on symptom duration, safety netting, self-care, and antibiotics. Patients felt confident to self-care and were averse to consulting with HCPs unnecessarily but struggled to assess symptom severity. Patients reported seeking help for children or elderly dependents earlier. HCPs’ concerns included patients’ attitudes and a lack of available monitoring of advice given to patients. Participants believed community pharmacy should be the first place that patients seek advice on common infections. The patient information leaflet on common infections should be used in primary care and community pharmacy to support patients to self-manage symptoms and determine when further help is required.

## 1. Introduction

Antimicrobial resistance (AMR) continues to be a global threat, largely driven by overuse of antibiotics [1]. Most antibiotics (70% in 2019) in England are prescribed in the community [2], with many inappropriate for self-limiting infections [3,4,5], and primary care continues to be a priority area for Antimicrobial Stewardship (AMS) programmes, to optimise antimicrobial use. Patient behaviour is a key target of the UK AMR national action plan (2019–2024); ambitions include preventing the need for antimicrobials and improving the publics’ infection prevention behaviours [6]. The National Institute for Health and Care Excellence (NICE) and Public Health England (PHE) recommend self-care actions as first-line treatment for many common infections [7] and recommend that HCPs use patient resources to educate patients on the management of common infections [8]. Among UK primary care clinicians, 94% report using patient-facing infection-related information leaflets [9]. Interventions to facilitate shared decision-making with patients, such as leaflets, are effective at reducing antibiotic use for acute respiratory tract infections in primary care [10]. Although there are patient leaflets covering specific or childhood infections, there are no general infection self-care leaflets for use in primary care and community pharmacies to facilitate consistent advice to patients, empowering patients to manage their current and future infections. 

Analysis of a large 2018/19 community cohort study in England found that most participants managed respiratory tract infection (RTI) symptoms without seeking medical attention, with only 14% leading to a GP or dental consultation and 10% antibiotic use [11]. The coronavirus disease 2019 (COVID-19) pandemic has seen an increase in GP telephone consultations, possibly resulting in more antibiotic prescriptions [12,13]. Interventions developed using behavioural science can improve antibiotic prescribing [14], and patient educational interventions developed using the TDF have high acceptability and feasibility [15,16]. A person-based approach to intervention development uses a range of qualitative research methods, evidence, and theory and puts end users at the centre of development in order to effect behaviour change [17]. 

This study aimed to explore patients’ and HCPs’ needs for providing or receiving infection self-care and safety-netting advice. Qualitative interviews and focus groups (FGs) informed by the TDF, and online questionnaires were used in this person-based approach, to inform the development of a patient and carer facing leaflet covering common infections.

## 2. Results

### 2.1. Participant Characteristics

A total of 27 patients and 7 HCPs participated in interviews and FGs and a further 5 HCPs and 25 patients (identifying from minority ethnic groups) provided feedback on a near final draft of the leaflet via questionnaires (Table 1). 

### 2.2. Emerging Themes on Management of Common Infections

#### 2.2.1. Patients

Patient themes regarding self-managing common infections are summarised in Table 2 (detailed in Appendix A).

(a) Preventing and self-treating infections

Patients believed that a healthy lifestyle could prevent infections and that handwashing could stop infections from spreading. Some believed that family upbringing influenced hygiene behaviour and that if you grew up in a house ‘too clean’, you were more susceptible to infections. Patients interviewed at the start of the COVID-19 pandemic reported more proactive health behaviours (e.g., oral hygiene) due to a perceived lack of healthcare access and preventing COVID-19 by following social distancing and increased cleaning. 

Patients reported self-care behaviours for common infections they believed their body could usually fight off, including common RTIs. Patients believed rest and hydration would help them to get better quicker. Some used over-the-counter (OTC) remedies, whilst others believed that it was important for symptoms to run their course rather than ‘suppress’ them with medicines and that taking any medicine too often could make them less effective. They were motivated to self-care and continued to carry out professional and social responsibilities. Behaviours were influenced by family traditions, previous successful methods, and advice from HCPs or the NHS website. Patients felt confident self-caring for familiar symptoms (e.g., cough) but less-so for perceived serious symptoms (e.g., rashes). 

(b) Health-seeking behaviour

Patients reported difficulty in assessing when to consult and were guided by concerns of potentially serious or persisting symptoms. Patients’ decision to consult was influenced by work, children, social pressure, and symptom tolerance. Failure to resolve symptoms through self-treatment was also a factor. Many patients expressed avoidance for seeking GP help, due to feelings of guilt about putting pressure on the NHS and their GP’s time, as a habit or family culture (‘we don’t tend to visit the GP’), or due to inaccessibility of appointments. A minority perceived that they accessed healthcare more than necessary due to anxiety about symptoms progressing to serious illness. When booking an appointment, patients reported negative feelings about discussing symptoms with reception staff. Patients reported a lower threshold for seeking help for children or elderly family due to concerns about vulnerability, uncertainty about how the infection might progress, and an expectation that they need antibiotics due to a weaker immune system. 

Many patients reported contacting community pharmacy staff for infection advice; their knowledge was trusted, and they were more accessible than GPs. Patients reported that public campaigns had increased their awareness of pharmacy services; a minority viewed pharmacies as a place to collect their medicine only. Some would not discuss certain symptoms in the pharmacy setting and were unaware of pharmacy consultation rooms, which are available in some pharmacy settings. 

(c) Healthcare expectations

Expectations for the consultation included wanting to have input into decisions, a rapid solution to their symptoms, and the reassurance that their infection would resolve. Patients would also consult when they believed antibiotics were required to treat an infection, including urinary, chest, ear and skin infections, and tonsillitis. Patients were familiar with appropriate antibiotic use messages from public campaigns, but had less knowledge of antibiotic resistance; however, they believed that if they took antibiotics too much they would not work. If antibiotics were prescribed, they viewed it as important to take them as their HCP directed. 

#### 2.2.2. Healthcare Professionals

HCP themes regarding provision of advice to patients with common infections are summarised in Table 3 (detail in Appendix A).

(a) Responsibilities and approaches to managing common infections

Overall, HCPs were motivated and optimistic that provision of self-care advice could help to prevent future infections and enable patients to self-care before seeking help and that these actions could reduce use of services and antibiotics. GPs discussed the benefit of shared decision-making and the importance of identifying expectations at the start of consultations and counselling against antibiotics, if not appropriate. Both GPs and pharmacists believed that community pharmacies were accessible and should be viewed as a patient’s first place for advice on common infections. Pharmacists reported some hurdles to providing patients with advice, which included not being able to see the patient’s record or indication and other responsibilities preventing them from having longer discussions.

GPs reported using AMS tools such as the TARGET patient leaflets [18] and reflected that they helped to reassure patients. Some used back-up prescriptions and appointments for safety-netting. A GP interviewed at the start of the COVID-19 pandemic briefly commented on the use of telephone consultations and expressed concern that this may lead to more antibiotics being prescribed. All HCPs reported that there were no systems in place to monitor advice given in community pharmacy or GP settings. For GPs, variations in antibiotic prescribing patterns at their practice were a concern and consequently their efforts to change patient’s expectations could be undermined by other prescribers.

(b) Patient attitudes and context

GPs and a nurse practitioner discussed patient expectations for antibiotics as a hurdle to overcome in consultations, particularly those patients who had been prescribed them for previous infections and those consulting for children. HCPs generally believed that patients’ understanding of AMR had improved somewhat in recent years due to public campaigns. In contrast, they believed that patients had less awareness of the importance of self-care but believed that campaigns and social media messaging could be effective at influencing patient behaviour. 

HCPs reported difficulty in knowing whether their advice had changed patients’ behaviour. There was some concern from GPs about a minority of patients who may immediately take the back-up antibiotic prescription or seek another prescriber to request antibiotics. HCPs reported barriers for patients to self-care for infections, including health inequalities and affordability of OTC medicines, literacy or language barriers, complex patients who are home bound, and those unable to take time off work.

### 2.3. Development of the Managing Common Infections leaflet

All participants from FGs, interviews, and questionnaires (Table 1) commented on iterations of the information leaflet. 

(a) Usefulness of leaflet covering common infections

Overall, participants across the study reported that a resource was required to educate patients about managing common infections. All HCPs responding to the questionnaire reported they were likely or very likely to use the leaflet with patients; HCPs rated every section of the leaflet as very useful or useful, except for a section which included statistics on which infections were most common in the population (80%, 4/5 rated useful); this section was subsequently removed. Among the minority ethnic patients responding to the questionnaire, 88% (22/25) reported the leaflet provided all the necessary information they would need to self-manage a common infection and all agreed that the content was useful or very useful.


*‘Great way to get a lot of information to the patient. Will lead to less calls.’*
(nurse, questionnaire)


*‘I think it’s a very good leaflet, if the idea of this is to actually reduce the use of antibiotics when other simpler remedies could be used then I think it’s a good idea.’*
(Male patient, Focus Group 5)

(b) Information priorities for common infection leaflet

Priorities were similar across all participants and included information on when a GP appointment was needed, serious symptoms to look out for, average duration of illness, advice on self-care and OTC medicines for symptoms, and advice on when antibiotics are needed. Information needed to be simple with website links to further advice. Patients in FGs requested information on signs of sepsis and information to help distinguish their infection from COVID-19. Patients highlighted the need to clarify the difference between existing chronic illnesses which may present similarly to symptoms of common infection; for instance, for a urinary infection, symptoms may include passing urine more often at night. Patients also highlighted the need to include information on social distancing to protect vulnerable family members from infections.


*‘Many BAME [Black and Minority Ethnic] patients have other underlying issues, which may present the same symptoms as described in the leaflet, maybe expressing if there is some difference to the normal symptom they present?’*
(patient, questionnaire)


*‘I think if it could be quite basic if it has links to things on the internet…and then if you need to know more, this is where you go and look… and that set you on the path to helping yourself by finding out other things online.’*
(Female patient, Interview 1)

(c) Leaflet Design

Preferences from participants in interviews and FGs were for the leaflet to follow a logical step-by-step process for managing infections, with subtitles consisting of questions on personal actions patients could take, which could be discussed with an HCP. Suggested amendments from participants included the use of plain English, a reduction in the quantity of text, and the use of more images/icons to help understanding. Minority ethnic patients highlighted that too much text was inaccessible and intimidating, especially for patients where English was an additional language.

The final leaflet titled ‘How can I manage my common infection?’ (Figure 1) followed behavioural steps which helped the reader to make decisions on how to manage their own infection:What are the symptoms of a common infection?What if I think I have coronavirus (COVID-19)?How can I treat a common infection?How long could my infection last?Will my infection need antibiotics to get better?How can I stop my infection from spreading?What symptoms of serious illness should I look out for?What if I suspect signs of sepsis?


*‘[For the title] I want to say, how you can manage your common infection, question mark… I think it will put the emphasis on empowering the person. Them at the centre of this, because that’s what I think it should be about.’*
(Pharmacist, focus group 1)

(d) Suggestions for implementation of leaflet

HCPs believed the leaflet would be useful as a shared decision aid to support their discussions with patients and help to educate patients in the GP and community pharmacy settings. Patients reported they would expect to receive the leaflet from a GP or pharmacist. All participants reported preferences for both printed physical and online versions of the leaflet; a GP acknowledged the latter would be useful for remote consultations. The leaflet was published as part of the TARGET antibiotics toolkit [18], including an online web version and has been translated into 26 different languages, following feedback from minority ethnic patients.


*‘Depending on where they receive the leaflet. It could be attached to their prescriptions when they collect it from the pharmacy, or handed out at the GP surgery.’*
(pharmacy staff, questionnaire)


*‘Unless it is translated, this would be a lot of text for an average non-English speaker. Visuals are great on this but I think there would be more required to help them better understand.’*
(patient, questionnaire)

## 3. Discussion

### 3.1. Summary

Patients were confident about self-managing most common infections. Patients felt guilty about consulting GPs, preferring to self-care with OTC medicine, fluids, and rest unless symptoms became serious; however, they held contrasting expectations for children and elderly family members. Patients’ decision to consult was based on perceived severity and concern for duration of symptoms or a belief that their symptoms were not improving through self-care; however, they struggled to judge when common infections were serious. HCPs reported patient’s attitudes and expectations for antibiotics as their main hurdle to overcome in consultations where antibiotics were not necessarily needed, sometimes impacted by contradictory prescribing or advice from other prescribers. However, HCPs noted there were no systems in place to monitor patient advice, and they did not always know if their advice had influenced patients’ behaviour. Both patients and HCPs valued the importance of self-care, indicated community pharmacies should be used first for advice, and recognised the need for reassurance and a focus on safety netting in consultations. Participants from the interviews, FGs, and questionnaires reported that a general infection information leaflet was useful and needed, and their priorities and feedback were used to finalise the content and design of the leaflet. 

### 3.2. Strengths and Limitations

This study used a rigorous person-based approach [17] to develop a patient information leaflet for use in primary care and community pharmacy, underpinned by behavioural theory [21] throughout interview question development, analysis, and intervention development. Triangulation of data from patients and HCPs via multiple methods led to a large sample of participants contributing to the leaflet development, with similar themes across information and design needs.

The leaflet was designed to include general information to support HCPs’ provision of advice to patients; a strength is that the leaflet may be applicable in many situations and useful when the indicated illness is not known (this is quite common in the pharmacy setting [22]). A limitation is that the leaflet cannot provide the specific advice required for some patients—for instance, those with existing conditions, where safety-netting thresholds may differ; in these contexts, the leaflet may not be suitable or HCPs may need to augment the advice provided in the leaflet. 

There is potential for researcher bias in qualitative studies where preconceptions about the beliefs of participants could influence the data collection and analysis. We believe this effect was mitigated because multiple researchers collected and analysed data and met often to reflect on their role in the research and because a steering group with a range of backgrounds was involved in the design of interview schedules and interpretation of themes to inform the leaflet. 

Part way through the study, interviews and FGs had to be conducted remotely and participants were encouraged to have their camera on to allow the interviewer to pick up on visual cues. The researchers consider that the input from participants and quality of data was similar with both methods.

The recruitment strategy relied on participants volunteering and may be subject to sampling biases, which could prevent transferability to other populations; however, the use of several recruitment strategies reduced this, including recruitment of minority ethnic patients. Likewise, themes may not be transferable to specific patient populations, including those with pre-existing conditions. There were a larger number of females participating in interviews/FGs than males, which is common in qualitative studies in this area [23,24,25,26,27] and may relate to volunteer bias; however, there was a more equal ratio in the questionnaire participants. Research studies often have low participation of minority ethnic groups [28], and where these involve designing interventions, it can reduce usability across different groups. We invited minority ethnic patients participating in another related study to complete the online questionnaire; potentially this method may have been a barrier for patients lacking access to the technology; however, we achieved views from patients identifying from a diverse range of ethnicities. 

A limitation was not recruiting patients based on education/literacy level and socioeconomic status, as these aspects may affect accessibility of the leaflet. However, the leaflet was reviewed by the Plain English campaign (crystal mark 23499), and we recommend that HCPs use the leaflet in conjunction with their own advice to account for the context of patients. 

### 3.3. Comparison with Existing Literature

In our study, patients reported being more likely to seek help for suspected infections where they expected antibiotics, one of which was UTIs, and this aligns with previous cross-sectional research where 95% of women with UTI symptoms consulted an HCP [29]. A qualitative study exploring women’s process of self-care to GP consultation for UTIs described similar health-seeking behavioural influencers to our study, including perceived long duration and severity of symptoms [30]. Similarly, for RTIs, patients judged severity of symptoms on an evaluation of symptom duration, and their self-care actions were influenced by advice from their GP, pharmacy, and media [31]. In an evaluation of an AMS intervention to support community pharmacy staff advice about common infections, 20% of patients reported they did not know how long it would take them to feel better [32]. Of the literature on parents’ expectations for children with common infections, a key similarity to our study was a belief of parents that consulting and being prescribed antibiotics were viewed as safer options [33,34]. Parents reported uncertainty about when to seek help for children and how to judge the seriousness of symptoms [35,36], and studies echo our implications that support is needed for patients around decision-making and self-care. 

HCPs’ concerns around patients’ expectations for antibiotics align with previous research [37]; however, patients in our study had better knowledge of appropriate antibiotic use in comparison with the qualitative literature [23,24,25,26,27,38,39,40], which was conducted mainly before 2012. Since 2012, there has been greater emphasis on public education via national AMR campaigns, support for GPs with AMS tools [18], and incentives for Clinical Commissioning Groups around improvements in antibiotic prescribing (Quality Premiums) [41], which may have caused this attitude shift. Some patients in our study held the belief that home cleaning practices could affect immunity, which has been reported in recent public surveys [42,43]. Messages around targeted hygiene, in line with COVID-19 messaging, may help to improve these misconceptions [44]. The development of a general self-care leaflet is supported by findings of systematic reviews of shared decision-making (SDM) which have found that decision aid interventions for acute RTIs in primary care can improve patient knowledge and satisfaction and can reduce antibiotic prescribing [10,45]. HCPs concerns around the prescribing or provision of contradictory advice of other prescribers in their practice is common in the literature [16,37] and may be helped through practice-based audits and regular training. A new finding is HCPs concern around the lack of systems and processes to monitor the self-care advice given to patients, and further work may be needed to explore how this can be embedded into existing auditing. 

In our study, participants views on community pharmacy concur with the literature with pharmacy staff, who view educating patients as one of their key AMS roles [22]. An evaluation of the TARGET UTI leaflet found that 25% of patients with UTI symptoms had sought help first from a pharmacist, and 65% were comfortable discussing their symptoms in private settings with a pharmacist [46]. Evaluation of a patient-facing RTI leaflet in community pharmacies led to a decrease in GP referrals for RTIs and an increase in provision of self-care advice in the intervention group [47]. This supports the use of the general infection leaflet in pharmacies, but research should explore the impact of community pharmacy staff’s advice using these leaflets on patients’ AMS behaviours.

### 3.4. Implications for Research and Practice

Patients have inherent views, expectations, and traditions around managing common infections, and HCPs should start with an honest dialogue to gauge a patient’s expectations and understand their unique context. The managing common infections leaflet developed from our study can support shared decision-making and patient education in the GP and pharmacy setting, which may reduce unnecessary consultations and antibiotic use [48]. As a general information leaflet, we recommend that the patient’s specific context and needs should always be considered and that appropriate tailored advice provided in addition to the leaflet as necessary. The managing common infections leaflet is freely available via the TARGET toolkit (available on the Royal College of General Practitioners (RCGP) website [18]) and will be reviewed against guidance regularly. 

Patients in this study believed antibiotics were required to treat ear infections and tonsillitis, most of which are self-limiting. In a 2020 public survey of 2022 people, over two-thirds (68%) incorrectly stated that antibiotics are needed to treat most ear infections [49]. Antibiotic treatment for most ear infections opposes NICE guidelines, and it is therefore an important area for further research to understand the source of these expectations. Although we involved minority ethnic patients in the design of the leaflet, to ensure information and advice is reaching those who need it most, further work is needed with diverse minority ethnic groups and high deprivation communities to explore the accessibility and affordability of treatment for self-managed common infections and to implement and evaluate the effect of AMS educational interventions and patient-facing infection leaflets. 

Implementation of patient-facing resources will become increasingly important to support carers and patients with common infections as COVID-19 social distancing measures reduce. Remote consultations are anticipated to continue [50], and future research should explore implementation of patient information leaflets in this context and evaluate their effect on antibiotic prescribing. The COVID-19 pandemic is likely to influence the themes around patient management of infections reported in our study, and further work is needed to explore the impact of the pandemic on public health-seeking behaviour and attitudes towards antibiotics and hygiene.

## 4. Materials and Methods

### 4.1. Study Design

Qualitative study using FGs, interviews, and online questionnaires with patients and HCPs to inform parallel development of a patient leaflet covering management of common infections. The qualitative schedules (Appendix A) for HCPs and patients included two sections: the first covered approaches to managing common infections, and the second involved reviewing and giving feedback on the leaflet under development. Questions were informed by a literature review, expert stakeholder input, and the TDF [21] in order to ensure questions covered all behavioural determinants. Iterations of the leaflet were reviewed in FGs and interviews, and participants were encouraged to think-aloud their immediate views on the information and design. Following completion of interviews and FGs, the research team decided to collect additional data from HCPs and specific patient groups via online questionnaires to finalise the leaflet. Questionnaire participants reviewed a near-final version of the leaflet towards the end of data collection. Questionnaires (Appendix A) were hosted on the PHE Select Survey platform and were mostly qualitative open questions, with a small number of quantitative Likert-type scale questions on usefulness and completeness of information covered in the leaflet. 

The research team and study steering group (including patient and HCP representatives) were experienced in qualitative research, behavioural science, and intervention development for the primary care setting. The Consolidated Criteria for Reporting Qualitative Research (COREQ) was observed in this report (Appendix A).

### 4.2. Intervention Development

#### 4.2.1. Audience and Purpose of the Leaflet

The study aimed to develop a patient leaflet including general information and recommendations around common infections that could support HCPs with provision of advice to adult patients in primary care consultations and in community pharmacy settings. The leaflet was not intended to replace HCP advice, but to support it. As a general leaflet, it was not designed to account for specific patient needs, for instance, those with existing conditions who would likely need specific tailored advice from their HCP. 

#### 4.2.2. Leaflet Development

An initial leaflet draft incorporated evidence from a review of the literature on patient self-care actions for common infections, incorporating recommendations from NICE and PHE guidance and from stakeholder reviewers (different from participants in this study). The stakeholder reviewers included a range of healthcare professionals, infectious disease experts and consultants, and professionals with a background in behavioural science and knowledge and literacy. Stakeholders were identified and recruited from PHE and from a range of clinical groups and networks. Stakeholders reviewed and provided input on iterations of the leaflet throughout the study. In line with the person-based approach to intervention development [17], a log of suggested changes were kept, and leaflet drafts were revised iteratively, with support from a design team. After questionnaire feedback was incorporated, the leaflet was finally reviewed by a Plain English group to fine-tune language, and stakeholder reviewers approved final content before publication.

### 4.3. Setting and Participant Recruitment

Participants were recruited January–May 2020 through a range of methods. For HCPs participating in interviews and FGs, invitations were cascaded through the Gloucestershire Clinical Commissioning Group (CCG), existing networks, mailing lists, and social media. Recruitment was purposive and aimed to recruit GPs, community pharmacists, and nurse representatives from across England. For online questionnaires, HCPs were invited to give feedback on the leaflet draft in May 2020 through the TARGET antibiotics toolkit mailing list and newsletter (sent to 750 individuals).

Patients participating in interviews and FGs were recruited through the community and known networks, including a PHE public representatives’ network. Inclusion criteria for patients included being over 18 years, fluent in English, and not suffering from long term illnesses or co-morbidities, as these patients may have complex information needs that could not be captured in a general infection leaflet. Patients representing diverse minority ethnic groups, participating in another study led by members of the research team, were opportunistically invited to provide feedback on the leaflet via an online questionnaire in May 2020. Minority ethnic patients (*n* = 76) were recruited to this other study via purposive snowball and community sampling and were given the option to participate in an online questionnaire for the present study. 

Demographic data collected from all patients included self-reported age, gender, and ethnicity. Information on socioeconomic status and health literacy and education level were not collected.

### 4.4. Data Collection

Initially, semi-structured interviews and FGs were conducted face-to-face in Gloucester and London; following the introduction of COVID-19 restrictions, they were undertaken remotely on Skype. Type of data collection was dependent on participant preference and availability. Participants provided written and informed consent and had the opportunity to ask questions before data collection. Discussions lasted between 45 and 80 min for the initial questions covering management of infection and between 30 and 60 min for leaflet discussion, with a break in between; FGs tended to be longer in duration than individual interviews. The leaflet was printed for face-to-face participants, and it was emailed to Skype participants, ahead of the second section of the discussion. 

Data were collected by three female researchers (B.M., C.V.H., R.A.) experienced in qualitative methods and without a clinical background. Researchers made detailed field notes about participants’ feedback on the leaflet. Participants were not known to researchers prior to data collection and were paid for their time according to PHE policy on public involvement in research. Data were collected until researchers agreed that data saturation had been reached [51,52], where there were no new emerging themes. Discussions were recorded, transcribed by an external agency, and checked for accuracy by researchers. Recordings or transcripts were not returned to participants. All files were handled in accordance with the Data Protection Act 2018 and General Data Protection Regulations (GDPR).

### 4.5. Data Analysis

Inductive thematic analysis of qualitative data followed the six stages outlined by Braun and Clarke [53]. Data were analysed in full by one researcher (C.V.H.), and 20% were double coded by a second researcher (E.S.). First, researchers became familiar with the data, and NVivo Pro-11 software was used to organise the analysis. Researchers labelled transcripts with codes, meanings, and patterns emerging from the data. Codes were subsequently arranged into themes and iteratively revised as the dataset was reviewed against the coding framework. Researchers met regularly to discuss emerging themes and insights and to discuss their own beliefs and involvement in the research process; any conflicts in coding were resolved through discussion. Following agreement of a thematic framework by the research team and steering group, the themes were mapped to the TDF for reporting, and quotes which best illustrated the meaning of the theme were identified. The quantitative Likert-type scale questionnaire data were analysed descriptively using Microsoft Excel software. 

For leaflet development, two researchers collated the questionnaire findings and the field notes, emerging themes, and think-aloud feedback from the interviews and FGs. Following each data collection, key findings and suggestions were tabulated in Microsoft Excel, and discussion between the research team and steering group agreed changes or implications for the design, content, and implementation of the leaflet. This log of changes was reviewed iteratively to allow rapid changes to the leaflet ahead of subsequent interviews and FGs. 

Themes from the interviews and FGs about approaches and experiences managing common infections are presented in the Results Section 3.2, while specific feedback on the leaflet (including questionnaire data) are presented in the Results Section 3.3. 

## 5. Conclusions

A general patient information leaflet on common infections should support patients to self-manage symptoms, utilise community pharmacies for advice and empower patients to determine when GP consultation is required, especially for elderly or child dependents.

## Figures and Tables

**Figure 1 antibiotics-10-01113-f001:**
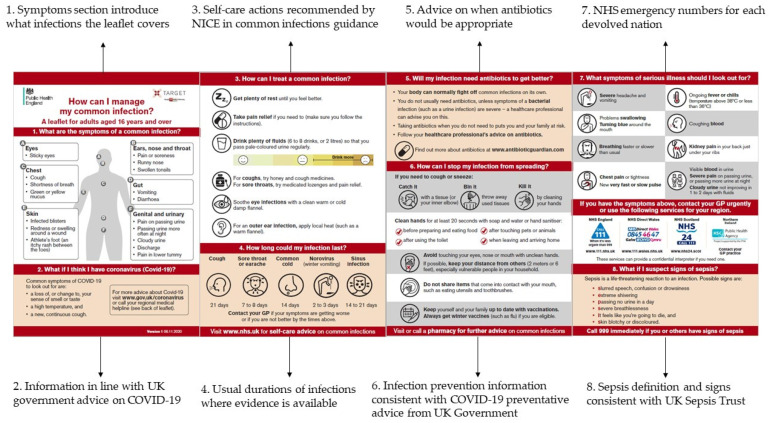
Labelled sections of the ‘How can I manage my common infection?’ leaflet. COVID-19—Coronavirus disease 2019. UK Government information on COVID-19 [19]. NICE—National Institute for Health and Care Excellence common infection guidance [8]. NHS—National Health Service. UK Sepsis Trust [20].

**Table 1 antibiotics-10-01113-t001:** Patient and healthcare professional self-reported characteristics.

Patients *n* = 52			Healthcare Professionals *n* = 12		
	INT and FGs (*n* = 27)	Questionnaire (*n* = 25)		INT and FGs (*n* = 7)	Questionnaire (*n* = 5)
**Age**			**Profession**		
16–24	1	3	Community	3	-
25–34	3	12	pharmacist		
35–49	8	1	General	2	-
50–64	8	2	Practitioner		
65+	7	5	Prescribing advisor	1	1
Unknown	-	2	Nurse practitioner	1	4
**Gender**			**Gender**		
Female	18	13	Female	3	-
Male	9	11	Male	3	-
Unknown	-	1	Unknown	1	5
**Ethnicity**					
Bangladeshi	1	5			
Black African	-	2			
Black Caribbean	1	2			
Chinese	-	5			
Indian	-	6			
Pakistani	1	2			
Sri Lankan	2	1			
White British	20	-			
White European	2	2			

INT—Interviews. FGs—Focus Groups.

**Table 2 antibiotics-10-01113-t002:** Emerging themes of patients on self-management of common infections.

Theme	Sub Themes	TDF Domain(s)	Quotes	Implications for Leaflet
Preventing infections	Healthy lifestyle Motivations for preventing infections Social distancing and hygiene	Knowledge; Belief about consequences; Social influences; Reinforcement	‘I’ve suffered from urinary tract infections so drinking plenty of water is really important for me and having a lot of vegetables and fruit.’ (Female, FG5) ‘A lot of cleaning, a lot more cleaning. I used to clean a lot anyway but since this COVID-19, I find that it’s just in my head all the time.’ (Female, FG4)	Advice on preventing infections Focus on protecting vulnerable family members
Self-caring for infections	Self-care actions, attitudes, and motivations Perceived skills and confidence Contextual barriers	Skills; Belief about consequences; Belief about capabilities; Environmental context	‘Not being a child, you come across things that you’ve had in the past and you either using the tried and tested that you’ve done before… And if it is something that persists or goes on for longer, I go to the pharmacist.’ (Male, FG4)	Self-care advice for specific symptoms Signpost to NHS website
Health-seeking behaviours	Judging seriousness of symptoms Deciding to seek help Triaging and accessibility Children and elderly Community pharmacy	Memory attention and decision-making; Belief about consequences; Environmental context; Emotions	‘Anything where I’m feeling this is not just an easy cold to manage. This could have quite an impact on other people, because I’m self-employed as well. So it’s a really hard judgement call…’ (Female, FG2) ‘I think after COVID I start to get a bit more anxious now thinking is it something more serious and I think if I had more of a cold now, I’d probably seek more medical attention…’ (Female, FG5)	Advice on duration of symptoms and when to seek help Serious signs of illness Advice on use of community pharmacy
Healthcare expectations	Experience receiving infection advice Expectations for consultations Beliefs about antibiotics	Reinforcement; Goals; Knowledge; Belief about consequences; Intentions	‘Hopefully a way to end this illness. I don’t want to come out thinking I’m none the wiser than what I was before. I’d like to know that there’s an end in sight.’ (Male, FG3) ‘That belief that if you take antibiotics too frequently then they don’t actually work as well. That’s always been drummed into me, don’t take antibiotics for everything.’ (Female, FG3)	Advice on when antibiotics could help an infection

TDF—Theoretical Domains Framework. FG—Focus group.

**Table 3 antibiotics-10-01113-t003:** Emerging themes of healthcare professionals on provision of advice to patients with common infections.

Theme	Sub Themes	TDF Domain(s)	Quotes	Implications for Leaflet
Roles and responsibilities	Motivation for AMS and promoting self-care Professional responsibilities	Belief about capabilities; Optimism; Knowledge; Skills; Social and professional role	‘I’m optimistic in terms of what I can do to educate the patients, whether than then translates into a reduction in resistance is another matter. But I’m certainly confident in what I do.’ (INT 4, GP, Male)	Leaflet would support HCPs in provision of infection advice
Approaches to managing common infections	Importance of self-care advice Shared decision-making Encouraging use of community pharmacy Approaches to educating patients Variations in practice and lack of monitoring	Belief about consequences; Belief about capabilities; Goals Social influences; Environmental context	‘I personally place a huge emphasis on self-care because in this day and age of consent, shared decision-making, empowering the patient as well, culturally we’ve moved away from being told by healthcare professionals what to do and how to do it exactly. It is very much a collaborative process.’ (FG1, pharmacist, female) ‘[about remote consultations] …can’t physically assess their illnesses and [GPs] will probably prescribe more than if they were able to have that face-to-face physical assessment just to err on the side of caution unfortunately.’ (FG1, pharmacist, female) ‘I worked in a practice which had six partners plus extra doctors and the variation in the threshold for prescribing was enormous… I think if you’ve got that variation at a clinician level it’s very difficult to expect staff to have consistent messaging.’ (FG1, GP, Male)	Self-care advice for common symptoms Promote leaflet as a shared decision tool Community pharmacy advice
Patient attitudes and context	Beliefs about patient attitude/expectations Beliefs about effect on patient behaviour Mass media/public health campaigns Patient contextual barriers	Belief about consequences; Belief about capabilities; Environmental context; Social influences	‘It can be quite clear the patients might have a pathway in their mind about what the treatment should be like, for instance, rather than taking the information on board. I think that’s probably the biggest barrier, patient expectation.’ (INT 1, pharmacist, Male) ‘I have prescribed paracetamol in families that I know would have struggled… but you have to bear in mind that if they really are struggling financially, all the self-care advice that you give is going to be difficult if they can’t afford it.’ (INT 2, nurse practitioner, Female)	Information on when antibiotics are needed and consequences of using incorrectly Align to existing campaigns and resources (Keep Antibiotics Working, TARGET antibiotics toolkit)

TDF—Theoretical Domains Framework. FG—Focus group. INT—Interview. GP—General Practitioner.

## Data Availability

The thematic analyses generated during this study are included in full in this published article and its supplementary information files. The corresponding author can be contacted for further study materials on reasonable request.

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
