# Peer review of "Empowering Patients to Self-Manage Common Infections: Qualitative Study Informing the Development of an Evidence-Based Patient Information Leaflet"

_antibiotics, 2021, doi:10.3390/antibiotics10091113_

Round 1

Reviewer 1 Report

This study uses qualitative methodology to explore patient’s and health care professionals needs for providing or receiving infection self-care and safety netting advice. This was then used to inform the development of a consumer leaflet covering common infections. This is an important area of research and this methodology has previously been used by this author group to develop a UTI leaflet, but remains original.  However, I am concerned that end product leaflet has significant issues with respect to providing sufficient information in a safe way.  I note that the leaflet was designed to be available in primary care settings such as GP or community pharmacist settings and I would recommend authors review leaflet with input from infectious diseases physician expert groups. 

Specific areas of improvement

  • P2 line 49 Consider adding statement and reference - Interventions to facilitate shared decision making to address antibiotic use have been demonstrated to be effective. (e.g.,Coxeter P, Del Mar CB, McGregor L, Beller EM, Hoffmann TC. Interventions to facilitate shared decision making to address antibiotic use for acute respiratory infections in primary care. Cochrane Database Syst Rev. 2015;2015(11):CD010907. Published 2015 Nov 12. doi:10.1002/14651858.CD010907.pub2). 
  • P2 line 92 Add further detail on review of literature and expert review noting my concerns on the information provided in leaflet.
  • P3 line 110 Further information on description of minority groups and whether online questionnaire format was a barrier
  • P4 Table 1 Is there any information on literacy of patients e.g., education level
  • P12 Figure 1. I have concerns regarding information in leaflet and alignment with Summary Antimicrobial Prescribing Guidance, Managing Common Conditions (updated May 2021) Study Reference 7.
    • Does the leaflet adequately cover conditions that are NOT/ or should have a low threshold for self-management?
      • Sepsis/ systemically unwell -> yes
      • Not for the following:
        • Those at High risk of complications -> insufficient information flags provided for immunocompromised hosts.
        • Also specific body systems e.g., infected blisters -> reactivation HSV is very different to shingles or primary HSV that warrant early medical assessment  e.g. redness or swelling around wound, if associated with pain or at a post-surgical site, early medical assessment warranted e.g., any UTI symptoms related to STD risks should not be self-managed.
      • Above is not a comprehensive list but some examples
    • What symptoms of serious illness should I look out for? I note the following are not on the list-
      • Headache/ neck stiffness / vomiting as a clinical triad is always a medical emergency
      • Inability to keep fluids down for x period…
      • This is not a comprehensive list but some examples.

Author Response

We would like to thank the reviewer for these very helpful comments and feel the manuscript has been improved. Please see detailed point by point responses (in bold) to the reviewer comments below, relating to the tracked changes in the manuscript document.

Reviewer 1 comments:

This study uses qualitative methodology to explore patient’s and health care professionals needs for providing or receiving infection self-care and safety netting advice. This was then used to inform the development of a consumer leaflet covering common infections. This is an important area of research and this methodology has previously been used by this author group to develop a UTI leaflet, but remains original.  However, I am concerned that end product leaflet has significant issues with respect to providing sufficient information in a safe way.  I note that the leaflet was designed to be available in primary care settings such as GP or community pharmacist settings and I would recommend authors review leaflet with input from infectious diseases physician expert groups. 

We thank the reviewer for their comments and hope that the responses to the individual points listed below help to alleviate concerns around the content of the leaflet.

  1. P2 line 49 Consider adding statement and reference - Interventions to facilitate shared decision making to address antibiotic use have been demonstrated to be effective. (e.g.,Coxeter P, Del Mar CB, McGregor L, Beller EM, Hoffmann TC. Interventions to facilitate shared decision making to address antibiotic use for acute respiratory infections in primary care. Cochrane Database Syst Rev. 2015;2015(11):CD010907. Published 2015 Nov 12. doi:10.1002/14651858.CD010907.pub2). 

Thank you for this suggestion, the statement and reference has been added to the introduction.

  1. P2 line 92 Add further detail on review of literature and expert review noting my concerns on the information provided in leaflet.

Thank you for your comments, drafts of the leaflet were reviewed by a range of stakeholders, including infectious disease experts, throughout the study, and final approval for publication was provided from these stakeholders. We have clarified the stakeholder reviewers in the ‘intervention development’ section of the methods.

We appreciate your comments regarding the information in the leaflet, it was a delicate balance to include the most important information in an engaging way, and feedback from participants was to keep text in the leaflet to a minimum. We would like to further clarify the he purpose of this leaflet, which is to support HCP’s discussions with patients about managing common infections, and therefore includes general information and advice; a strength of this is that the leaflet may be applicable in many situations and useful when HCPs are uncertain of a patients indication (e.g. in the pharmacy setting). A limitation, as you have recognised, is that the leaflet cannot provide the specific and tailored advice required for some patients, for instance those with existing conditions or who are immunocompromised, where safety netting thresholds may differ; in these contexts the leaflet may not be suitable or HCPs may need to augment the advice provided in the leaflet. We have clarified the information above in the intervention development section of the methods, and in the limitations section of the discussion. We will also review the published user guide for the leaflet to ensure this information is prominent.

  1. P3 line 110 Further information on description of minority groups and whether online questionnaire format was a barrier

Thank you we have expanded on the demographics shared in table 1 to include those patients completing online questionnaires. We have also given more detail in the methods about these participants and included in the limitations a discussion around online questionnaires being a potential barrier for this population.

  1. P4 Table 1 Is there any information on literacy of patients e.g., education level

We did not collect this from participants and appreciate this is a limitation which we have added to the discussion. The leaflet was reviewed by the Plain English campaign to refine language.

  1. P12 Figure 1. I have concerns regarding information in leaflet and alignment with Summary Antimicrobial Prescribing Guidance, Managing Common Conditions (updated May 2021) Study Reference 7.
    • Does the leaflet adequately cover conditions that are NOT/ or should have a low threshold for self-management?
      • Sepsis/ systemically unwell -> yes
      • Not for the following:
        • Those at High risk of complications -> insufficient information flags provided for immunocompromised hosts.
        • Also specific body systems e.g., infected blisters -> reactivation HSV is very different to shingles or primary HSV that warrant early medical assessment  e.g. redness or swelling around wound, if associated with pain or at a post-surgical site, early medical assessment warranted e.g., any UTI symptoms related to STD risks should not be self-managed.
      • Above is not a comprehensive list but some examples
    • What symptoms of serious illness should I look out for? I note the following are not on the list-
      • Headache/ neck stiffness / vomiting as a clinical triad is always a medical emergency
      • Inability to keep fluids down for x period…
      • This is not a comprehensive list but some examples.

Thank you for your comments and we shall consider these for leaflet revision; as they do not directly relate to the manuscript, we will address them separately. This leaflet will also be reviewed against the updated NICE guidance from 2021.

In addition to this, we have added sections throughout the manuscript to clarify audience and purpose of this leaflet, and that it aims to support healthcare professionals in the advice given to patients, and that more tailored advice will be required for specific patients, which this leaflet may not cover. We appreciate the reviewer’s comments regarding the suitability of the information for certain patient populations and hope this alleviates their concerns.

Reviewer 2 Report

The public's campaign regarding to rational antibiotic use and self-management of common infections is important to help address the prevalent irrational use of antibiotics, which is partly driven by the populations. This study concerned a significant issue and aimed to explore healthcare professionals’ (HCPs) and patients’ needs on provision of self-care and safety-netting advice for common infections. The application of qualitative design and theory of TDF generates a lot of useful information to help us understand how to better educate the public. Though the decision making process of the public regarding to assessment of severity of symptoms, care-seeking behaviors and mis-concept of antibiotics is well-documented in existing studies. The development and feedback from HCP and the public on the leaflet is interesting and novel. The methodology and results is well-presented and relevant topic, especially the implementation and adoption of the leaflet by the public and further research (behavior change), is discussed. 

No revision comment was raised from my perspective.

Author Response

We would like to thank the reviewer for their comments about the manuscript and  appreciation of the importance of this work.

Reviewer 3 Report

  • Demography: Ethnic minority groups were considered during participant recruitment but no data given about the participant’s ethnicity. Is there any evidence that minority ethnic groups were actually in selected participants?
  • Education: Health education or participant’s education level plays an important role in answering these questionnaires. Participant’s education level is not mentioned in the research. For example, if a participant is a physician or a nurse practitioner their level of understanding common infections is different compared to an illiterate. So, depending on the education level different themes may arise. Please explain if researchers have done anything to mitigate this risk. If this is a limitation, please mention it in limitations section. 
  • Sex: Males and females’ number is different in study. Do you expect the results to be same if their number is same?
  • Please explain if there is any researcher bias in finding the emergent themes either from participants or HCPs. This is relevant based on conflicts of interest statement.
  • Limitations: Participants with no pre-existing conditions were chosen. Please explain if the results can be generalized to participants with pre-existing conditions.
  • Each theme can be complicated by COVID-19. Please explain the importance of COVID-19 in each theme, if the data is available.
  • Introduction is about antibiotic resistance. I believe the title will be contradictory to prevent antibiotic resistance if patients are empowered to self treat common infections with antibiotics. Is this data available and mentioned in the themes specifically? 
  • Limitations section can be improved as mentioned above.

Author Response

We would like to thank the reviewers for these comments and feel the manuscript has been improved. Please see detailed point by point responses (in bold) to the reviewer comments below, relating to the tracked changes in the manuscript document

  1. Demography: Ethnic minority groups were considered during participant recruitment but no data given about the participant’s ethnicity. Is there any evidence that minority ethnic groups were actually in selected participants?

Thank you we have expanded on the demographics shared in table 1 to include those patients completing online questionnaires. We have also given more detail in the methods about these participants and expanded on limitations.

  1. Education: Health education or participant’s education level plays an important role in answering these questionnaires. Participant’s education level is not mentioned in the research. For example, if a participant is a physician or a nurse practitioner their level of understanding common infections is different compared to an illiterate. So, depending on the education level different themes may arise. Please explain if researchers have done anything to mitigate this risk. If this is a limitation, please mention it in limitations section. 

Thank you for your comments, we did not collect this data from the interview or questionnaire participants and appreciate this is a limitation we have expanded on in the discussion. Patient participants were members of the general public and did not come from a healthcare background.

The leaflet was reviewed by the Plain English campaign and we recommend that the leaflet is used as an aide to support that advice that healthcare professionals give to patients and not replace it. Therefore, we recognise that healthcare professionals using this leaflet will need to tailor their information to patients depending on their context (e.g. literacy level, socioeconomic status). A completely pictorial version of the leaflet may be a great project to take forward in the future.

  1. Sex: Males and females’ number is different in study. Do you expect the results to be same if their number is same?

Thank you for your comment, this is a limitation we have expanded on in the discussion section. This is a possible volunteer bias and is common in existing qualitative studies in this area. There was a more equal ratio in the questionnaire participants as reflected in table 1.

  1. Please explain if there is any researcher bias in finding the emergent themes either from participants or HCPs. This is relevant based on conflicts of interest statement.

Thank you for this comment, we recognise in the limitations section that as the research team have worked within this area already i.e. developing interventions and exploring patient and healthcare professionals knowledge; that the may have been existing beliefs about the participants and what the findings may have been. However we used several strategies to mitigate this as recognised in the literature for qualitative studies: we stuck to a semi-structured topic guide which was developed with a steering group of individuals with different experiences and backgrounds, a number of researchers conducted the data collection and analysis and met regularly to reflect on their beliefs and impact on the research.

  1. Limitations: Participants with no pre-existing conditions were chosen. Please explain if the results can be generalized to participants with pre-existing conditions.

Thank you, we aimed to develop a leaflet with general information that could support healthcare professionals in the advice they give to patients. We therefore decided not to recruit patients suffering from long term illnesses or co-morbidities, as these patients may have more complex information needs that could not be captured in the general infection leaflet. We have added sections throughout the manuscript to clarify audience and purpose of this leaflet, and that the leaflet should not replace the advice provided by healthcare professionals but support it, and therefore more tailored advice will be required for specific patients, which this leaflet may not cover. We appreciate the reviewer’s comments about transferability of the themes to other patient populations and discuss this in the limitations.

  1. Each theme can be complicated by COVID-19. Please explain the importance of COVID-19 in each theme, if the data is available.

Thank you for this comment and we wholeheartedly agree. As the majority of the data collected in this study was before or at the start of the COVID pandemic, we cannot accurately describe any differences in this paper, however we have expanded in the discussion that further work is needed to explore patient health-seeking behaviours in light of the pandemic.

  1. Introduction is about antibiotic resistance. I believe the title will be contradictory to prevent antibiotic resistance if patients are empowered to self treat common infections with antibiotics. Is this data available and mentioned in the themes specifically? 

Thank you for your comment, the aim of the study was to investigate patient and healthcare workers approaches to managing common infections, and to develop a leaflet to help empower patients to self-care for symptoms of common infections (with rest, fluids and suitable over the counter remedies), and to help educate when a GP appointment may be necessary. We aim to improve use of antimicrobials and have reinforced the appropriate use of antimicrobials throughout the developed leaflet. Patients themes around antibiotics are described in full in the paper and there was no discussion around self-treating infections with antibiotics.

  1. Limitations section can be improved as mentioned above.

Thank you we have expanded the limitations section.

Round 2

Reviewer 1 Report

I agree that it is a delicate balance to ensure information is provided in an engaging way. The leaflet is appropriate in settings where it is used  to support HCP discussions on patient self-care of common infections rather than in place of HCP discussions. I thank the authors for addressing this in the revision.  

This manuscript is a resubmission of an earlier submission. The following is a list of the peer review reports and author responses from that submission.